# *FedMomentum*: Preserving LoRA Training Momentum in Federated Fine-Tuning

## Abstract

Federated fine-tuning of large language models (LLMs) with low-rank adaptation (LoRA) offers a communication-efficient and privacy-preserving solution for task-specific adaptation. Naïve aggregation of LoRA modules introduces noise due to mathematical incorrectness when averaging the downsampling and upsampling matrices independently. However, existing noise-free aggregation strategies inevitably compromise the structural expressiveness of LoRA, limiting its ability to retain client-specific adaptations by either improperly reconstructing the low-rank structure or excluding partially trainable components. We identify this problem as *loss of training momentum*, where LoRA updates fail to accumulate effectively across rounds, resulting in slower convergence and suboptimal performance. To address this, we propose *FedMomentum*, a novel framework that enables structured and momentum-preserving LoRA aggregation via singular value decomposition (SVD). Specifically, after aggregating low-rank updates in a mathematically correct manner, *FedMomentum* applies SVD to extract the dominant components that capture the main update directions. These components are used to reconstruct the LoRA modules with the same rank, while residual components can be retained and later merged into the backbone to preserve semantic information and ensure robustness. Extensive experiments across multiple tasks demonstrate that *FedMomentum* consistently outperforms prior state-of-the-art methods in convergence speed and final accuracy.

## 1 Introduction

Pre-trained large language models (LLMs) have demonstrated remarkable generalization to diverse tasks (Touvron et al., 2023; Achiam et al., 2023; Guo et al., 2025), and their adaptability enables strong performance on domain-specific tasks through task-specific fine-tuning (Brown et al., 2020; Chung et al., 2024; Schmirler et al., 2024; Yuan et al., 2024). However, high-quality public datasets are projected to become increasingly scarce in the near future (Ye et al., 2024). This scarcity is particularly severe in privacy-sensitive domains such as healthcare (Silva et al., 2019; Courtiol et al., 2019; Oldenhof et al., 2023) and finance (Liu et al., 2021; Lin et al., 2022), where data sharing is highly constrained due to legal and ethical concerns (Madiega, 2021).

By exchanging only model parameters while keeping the data locally, federated fine-tuning inherits the privacy-preserving properties of federated learning (FL) and emerges as a promising solution for distributed LLM fine-tuning (Qin et al., 2024; Yu et al., 2023; Zhang et al., 2023). However, full fine-tuning is computing- and memory-intensive and incurs significant communication delays in FL settings. Low-rank adaptation (LoRA) (Hu et al., 2022) offers a much more lightweight and communication-efficient alternative, and a growing number of recent studies have investigated its application in FL.

Existing LoRA-based federated fine-tuning methods face a fundamental dilemma: mitigating aggregation noise while preserving the structural expressiveness of the LoRA modules. On the one hand, directly applying vanilla parameter averaging (McMahan et al., 2017; Zhang et al., 2024), *i.e.*, performing separate aggregations for the upsampling matrix $A$ and the downsampling matrix $B$, inherently violates the additivity of model updates. This is because the LoRA structure approximates model updates via a low-rank composition, which is distorted when $A$ and $B$ are aggregated independently, resulting in noisy updates. On the other hand, methods that aim to eliminate aggrega-

tion noise often compromise the structural expressiveness of LoRA module. Since the value of the LoRA parameters directly determines the gradient direction (Meng et al., 2024), improper merging of LoRA-induced weight updates into the backbone (Wang et al., 2024; Yan et al., 2025), or freezing partial LoRA parameters (Sun et al., 2024) result in information loss within the LoRA space, which leads to both directional drift and diminished step sizes, disrupting the optimization trajectory. These limitations are largely overlooked in existing studies, yet they slow convergence and degrade model performance. We refer to this as *loss of training momentum* in federated fine-tuning.

To address this issue, we propose to utilize matrix decomposition and dimensionality reduction to reconstruct the LoRA structure and maintain the main direction after noise-free aggregation. Specifically, the aggregation results of the clients' *delta weights* (the product of local LoRA downsampling and upsampling matrices) can be decomposed by singular value decomposition (SVD) (Golub & Kahan, 1965) into different components. The top-$r$ *major components*, which capture the majority of the transformation energy, form a new LoRA representation with the same rank as previous rounds. By reconstructing the LoRA module from the *major components*, this approach preserves training momentum in LoRA structure, thereby maintaining consistent optimization directions across rounds. Furthermore, to maintain residual semantic information beyond the top-$r$ major components and avoid aggregation noise, the *residual components* (with rank $s$) are retained until their value reaches a desirable threshold and then merged into the backbone. The other *negligible components*, which contribute marginally to the overall representation, can be safely discarded.

Building upon our SVD-based aggregation scheme, we propose a novel federated fine-tuning framework, *FedMomentum*. In the initialization stage, the server distributes a shared backbone model and initialized LoRA modules to all clients. Then, in each communication round, each client trains the LoRA module on its own dataset and uploads the updated weights to the server. The server performs SVD-based aggregation, reconstructing new LoRA modules from the principal components, preserving the residual components, and sending both to the clients. The clients merge residuals into the backbone, and load the updated LoRA modules to prepare for the next round. This process continues iteratively until convergence.

To sum up, our main contributions can be summarized as follows:

- We are the first to identify and analyze the phenomenon of loss of training momentum during federated fine-tuning caused by inappropriate LoRA updating, which leads to suboptimal convergence.
- We propose a new algorithm, *FedMomentum*, that performs LoRA fine-tuning in a federated setting by updating the low-rank matrices using a momentum-aware SVD-based scheme. This design explicitly preserves update directions across rounds, alleviating the momentum loss problem.
- We conduct extensive experiments across multiple tasks, demonstrating that our approach consistently outperforms existing federated fine-tuning baselines in terms of convergence speed and final performance.

## 2 BACKGROUND & MOTIVATION

### 2.1 LoRA IN FEDERATED FINE-TUNING

Parameter-efficient fine-tuning (PEFT) has emerged as an effective approach to reduce the computational and memory overhead associated with fine-tuning large-scale pre-trained models (Ding et al., 2023; Han et al., 2024). One of the most popular methods, low-rank adaptation (LoRA) (Hu et al., 2022) freezes the backbone parameters of the original model, denoted as $W \in \mathbb{R}^{d \times k}$, and injects a pair of low-rank trainable matrices $A \in \mathbb{R}^{r \times k}$ and $B \in \mathbb{R}^{d \times r}$, where the LoRA rank $r \ll \min(d, k)$. The model is then updated as follows:

$$W' = W + \Delta W = W + BA, \tag{1}$$

where $\Delta W$ captures the task-specific adaptation. This design enables a significant reduction in trainable parameters while preserving model performance.

As LLMs are widely deployed across multiple clients with private, heterogeneous, and often sensitive data, federated fine-tuning becomes a crucial paradigm to enable collaborative adaptation while

Table 1: Taxonomy of representative federated LoRA fine-tuning methods. Only our method satisfies all three desired properties: communication efficiency, noise-free aggregation, and training momentum preservation.

| Method | Communication Efficiency | Aggregation Correctness | Training Momentum |
|---|---|---|---|
| FedIT (Zhang et al., 2024) | ✓ | ✗ | ✗ |
| FLoRA (Wang et al., 2024) | ✓ | ✓ | ✗ |
| FFA-LoRA (Sun et al., 2024) | ✓ | ✓ | ✗ |
| FRLoRA (Yan et al., 2025) | ✗ | ✓ | ✗ |
| **Ours** | ✓ | ✓ | ✓ |

preserving user privacy. Similar to centralized environments, LoRA-based federated fine-tuning methods have been proposed to address resource limitations, which allow each client to parameter-efficiently fine-tune and communicate only these low-rank matrices to the server.

Although recent methods improve communication efficiency by transmitting only LoRA parameters, they often struggle to jointly maintain *aggregation correctness* and *training momentum*. Naïve aggregation strategies introduce bias or noise due to the non-commutative nature of low-rank matrix multiplication. Meanwhile, improper handling of LoRA structures can hinder the optimization process and result in *loss of training momentum*.

As summarized in Table 1, most existing approaches fail to address both challenges simultaneously. FedIT (Zhang et al., 2024) performs separate aggregations for the low-rank upsampling matrix $A$ and the downsampling matrix $B$. However, this leads to a mathematical bias because $\sum B_i \times \sum A_i \neq \sum B_i \times A_i$, deviating from the global LoRA fine-tuning objective. FLoRA (Wang et al., 2024) attempts to achieve noise-free aggregation by stacking the local LoRA matrices and updating the local models with $BA$. Essentially, FLoRA directly merges all $\Delta W_i = B_i A_i$ back into the backbone and then reinitializes the low-rank matrices in subsequent rounds. This process disregards the previously learned low-rank structures and consequently slows down convergence. FFA-LoRA (Sun et al., 2024) aggregates and updates only the matrix $B$. While this approach avoids aggregation noise, freezing $A$ restricts the representation space of the update, thereby affecting its performance. FRLoRA (Yan et al., 2025) applies FedAvg to aggregate LoRA matrices and correct the noise by adding a residual term to the backbone, but it fails to preserve the principal directions of the weight updates within the LoRA modules. Moreover, since FRLoRA transmits full-size residual weights instead of low-rank updates, it lacks communication efficiency.

## 2.2 ANALYZING LOSS OF TRAINING MOMENTUM

Some previous studies have shown that the convergence speed is slow at the beginning after initialization of LoRA matrices (Meng et al., 2024). This slow convergence is due to the small or even zero initialization of the low-rank matrices $A$ and $B$:

$$\frac{\partial L}{\partial A} = B^\top \left( \frac{\partial L}{\partial Y} \right) X^\top, \quad \frac{\partial L}{\partial B} = \left( \frac{\partial L}{\partial Y} \right) (AX)^\top, \tag{2}$$

where $L$ is the loss function, $X$ is the input feature of the layer, and $Y = WX + BAX$ is the output.

This issue is further exacerbated in federated fine-tuning with LoRA structural compromises for noise-free aggregation. FLoRA (Wang et al., 2024) employs a merge-and-reinitialization strategy, where the LoRA modules are merged into the backbone and reinitialized each round. This strategy leads to information loss in LoRA representations, which causes the subsequent optimization direction and step size to deviate from the prior trajectory, resulting in slower convergence and lower final performance. Similarly, freezing of the initialized upsampling matrix in FFA-LoRA (Sun et al., 2024) also affects training for the same reason.

To validate the theoretical observation, we conduct a series of pilot experiments in centralized and federated fine-tuning settings. In the centralized settings, we investigate three strategies: 1) vanilla training of the low-rank matrices, 2) merging the low-rank matrices back into the backbone and then reinitializing them before subsequent updates, and 3) freezing of initialized upsampling matrix in LoRA module. Our results show that the reinitialization strategy and the freezing strategy not only slow the convergence at the outset but also eventually reach a sub-optimal performance, as illustrated in Figure 1(a). These results highlight the importance of preserving the structural expressiveness of

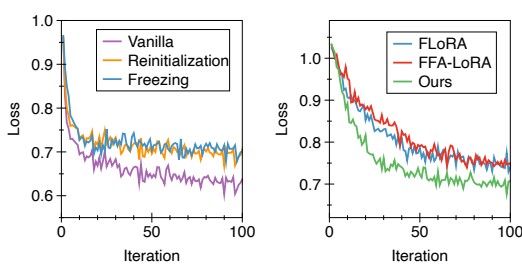

(a) Centralized Fine-tuning  (b) Federated Fine-tuning

Figure 1: Training loss curves of fine-tuning LLaMA2-7B on MetaMathQA with 10 clients under different LoRA update strategies. We compare various centralized fine-tuning and federated fine-tuning strategies under a unified setting: LoRA rank = 16, batch size = 16, local update steps = 10, and 100 iterations. The same experimental settings are adopted in Figure 1.

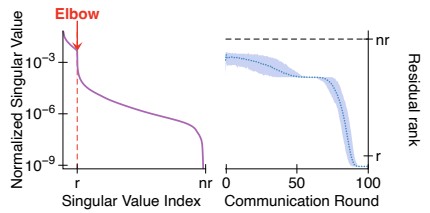

Figure 2: *(Left)* Normalized singular value spectrum for the first LoRA module of the first training round. The elbow point indicates the effective rank $r$ of the matrix. *(Right)* Residual rank statistics across all LoRA modules throughout training rounds. The solid line represents the average residual rank, while the shaded area reflects the range between the maximum and minimum observed values.

the LoRA module and minimizing information loss to maintain training momentum during fine-tuning.

Figure 1(b) illustrates that FLoRA and FFA-LoRA exhibit slower convergence and degraded final performance due to the *loss of training momentum*. In contrast, since our *FedMomentum* maintains the principal update direction, it effectively preserves the training momentum and achieves lower final training loss with faster convergence.

### 2.3 EMPIRICAL INSIGHTS OF SVD-BASED AGGREGATION

In federated fine-tuning, each client independently adapts the global model by computing local low-rank updates, $\Delta W_i = B_i A_i$. This ensures that each client's individual update has a rank of at most $r$. When these local updates $\Delta W_i$ are aggregated at the server to form a global update $\Delta W = \sum_{i=1}^{n} \Delta W_i$, the theoretical upper bound on the rank of $\Delta W$ is $nr$.

However, in Figure 2, our empirical investigations reveal that despite this theoretical maximum, the effective rank (Roy & Vetterli, 2007) of the aggregated $\Delta W$ still remains low. Specifically, when $\Delta W$ is decomposed using SVD, the number of *major components* is approximately $r$. While *residual components* with a rank of $s$ are observed, $s$ tends to continuously decrease as the federated fine-tuning progresses through rounds. This phenomenon is a strong indicator of LoRA's convergence, as the variation in client-specific updates diminishes in later stages of training.

The limited number of the effective components across different LoRA modules after aggregation underscores that the fundamental low-rank structure of LoRA is largely preserved, making it feasible to reconstruct new LoRA modules effectively while retaining critical information. This observation motivates our approach to utilize the major components for maintaining the representation capacity and the training momentum.

## 3 METHODOLOGY

### 3.1 OBJECTIVES & CHALLENGES

Our goal is to enable effective and efficient federated fine-tuning of LLMs by maintaining the training momentum and avoiding noisy aggregation. Specifically, we aim to achieve noise-free aggregation of low-rank updates while preserving the continuity of local adaptation through a novel SVD-based parameter updating mechanism.

However, the objectives are met with several non-trivial challenges:

**Inherent dilemma between noise-free aggregation and continuous adaptation**: As demonstrated in previous work, lossless merging of low-rank updates often disrupts the continuity of training (*e.g.,*

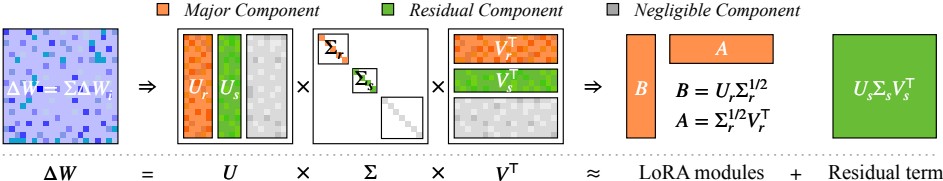

Figure 3: Overview of the SVD-based aggregation process.

due to reinitialization in LoRA), while methods that avoid reinitialization typically suffer from noisy or biased aggregation.

**Information loss induced by truncated SVD**: To maintain low-rank structures compatible with LoRA, retaining only the top-$r$ singular components of the aggregated update discards the residual subspace, resulting in noisy aggregation that may undermine model representation capacity.

**Gradient imbalance due to the singular value distribution**: When incorporating SVD into the aggregation process, the highly skewed magnitude of singular values can introduce anisotropy in the update directions, potentially affecting convergence stability and optimization consistency.

**Computational overhead of SVD**: Performing exact singular value decomposition (SVD) on large-scale model updates can be prohibitively expensive, especially in federated settings where communication and computation are constrained.

## 3.2 SVD-BASED LoRA AGGREGATION

To address these challenges, we propose an SVD-based LoRA aggregation algorithm that enables the structured, noise-free, and efficient aggregation of low-rank updates.

Given local updates $\Delta W_i = B_i A_i$ from all $n$ clients, we first aggregate them directly:

$$\Delta W = \sum \Delta W_i = \sum B_i A_i. \tag{3}$$

This aggregation respects the update additivity and avoids introducing noise through the separate averaging of $A_i \in \mathbb{R}^{r \times k}$ and $B_i \in \mathbb{R}^{d \times r}$. To continue training with low-rank structures, we decompose the aggregated high-dimensional $\Delta W$ into a low-rank approximation using truncated SVD, which allows us to reconstruct compatible LoRA matrices. Without loss of generality, we assume $k = \min(d, k)$ and formulate the decomposition as:

$$\Delta W \xrightarrow{\text{SVD}} U\Sigma V^\top = \sum_{i=1}^{k} \sigma_i u_i v_i^\top, \tag{4}$$

where $U = [u_1, \ldots, u_d] \in \mathbb{R}^{d \times d}$, $V = [v_1, \ldots, v_k] \in \mathbb{R}^{k \times k}$, and $\Sigma = \text{diag}(\sigma_1, \ldots, \sigma_k) \in \mathbb{R}^{d \times k}$.

Performing exact SVD on the aggregated update $\Delta W \in \mathbb{R}^{d \times k}$ incurs a computational complexity of $\mathcal{O}(d^3)$, which is prohibitively expensive for large-scale models. To accelerate the decomposition of the aggregated update $\Delta W$ while maintaining comparable approximation quality, we adopt a customized randomized SVD (Halko et al., 2011) adapted to our federated low-rank setting. We first generate a random Gaussian matrix $\Omega \in \mathbb{R}^{k \times c}$ and compute $Y = \Delta W \Omega$, where $c$ is the target sketch size that controls the accuracy of the approximation. Based on the structure of our aggregated update $\Delta W = \sum_{i=1}^{n} B_i A_i$, where each $B_i A_i$ is rank-$r$, the rank of $\Delta W$ is at most $nr$. Therefore, setting $c = nr$ suffices to capture all significant components of $\Delta W$ without loss of information.

We then obtain an approximate orthonormal basis $Q \in \mathbb{R}^{d \times c}$ by performing QR decomposition (Francis, 1961) on $Y$, *i.e.*, $Y = QR$.

Next, we project $\Delta W$ onto the low-dimensional subspace spanned by $Q$ and perform standard SVD on the resulting smaller matrix:

$$P = Q^\top \Delta W = \widetilde{U}\Sigma V^\top. \tag{5}$$

Finally, we approximate the left singular vectors of $\Delta W$ via $U = Q\widetilde{U}$.

We truncate the decomposition to the top-$r$ singular components to obtain a rank-$r$ approximation:

$$B = U_r \Sigma_r^{1/2}, \quad A = \Sigma_r^{1/2} V_r^\top, \tag{6}$$

where $U_r = [u_1, \ldots, u_r]$, $V_r = [v_1, \ldots, v_r]$, and $\Sigma_r = \mathrm{diag}(\sigma_1, \ldots, \sigma_r)$. This construction ensures that the product $BA$ closely approximates the aggregated update $\Delta W$, enabling high-fidelity reconstruction of the low-rank matrices for subsequent training rounds. By minimizing information loss within the LoRA module, it preserves structural expressiveness and helps maintain training momentum. Moreover, rather than using an unbalanced reconstruction (*e.g.*, $B = U_r \Sigma_r$, $A = V_r^\top$), each singular value is evenly split between $B$ and $A$ via $\Sigma_r^{1/2}$. The *balanced allocation* of singular values mitigates gradient imbalance in the following training iteration, which arises from the skewed singular spectrum and leads to instability or slow convergence in federated training.

The lower-energy components, associated with the residual subspace, are defined as:

$$W_{\text{residual}} = U_s \Sigma_s V_s^\top, \tag{7}$$

where $s$ denotes the number of residual components, $U_s = [u_{r+1}, \ldots, u_{r+s}]$, $V_s = [v_{r+1}, \ldots, v_{r+s}]$, and $\Sigma_s = \mathrm{diag}(\sigma_{r+1}, \ldots, \sigma_{r+s})$. The number of residual components $s$ is selected to ensure that the cumulative energy of the retained components (both major and residual) reaches a predefined threshold, such as 99% of the total energy. The residual term can be tracked for robustness or error correction and merged into each client's local backbone to mitigate noise during LoRA reconstruction.

The negligible components, which have minimal impact on the parameter space, can be discarded to reduce unnecessary computation and storage overhead.

### 3.3 *FedMomentum*: SVD-BASED FEDERATED FINE-TUNING

By employing the SVD-based aggregation algorithm, we propose *FedMomentum*, a four-stage federated fine-tuning framework that preserves LoRA training momentum without aggregation noise.

**Stage 1. Initialization of LoRA.** The server initializes the LoRA modules with a predefined rank and initialization strategy, and then distributes the backbone $W$ and the initialized LoRA modules to each client. As our focus is on aggregation rather than initialization, we adopt a default setting with a fixed rank and randomly initialized LoRA modules.

**Stage 2. Local Fine-tuning.** Each client trains the LoRA modules on its own datasets for $l$ local steps. The updated weights of LoRA modules are then sent back to the server.

**Stage 3. Aggregation and Reconstruction.** As the last section describes, upon receiving the LoRA modules from all clients, the server aggregates the local model updates without noise and utilizes randomized SVD to decompose the global model update into *major components*, *residual components*, and *negligible components*. The server reconstructs new LoRA modules for the next iteration from the *major components*. It then distributes the reconstructed LoRA modules and the corresponding *residual components* to all clients.

**Stage 4. Update Local Models.** Each client merges the residual components into the backbone and loads the new LoRA modules for the new round of training.

The federated fine-tuning process repeats stage 2 to stage 4 until convergence.

## 4 EXPERIMENTS

In this section, we present comprehensive experiments to evaluate the effectiveness and efficiency of *FedMomentum*. We benchmark its performance against existing federated fine-tuning approaches across diverse LLM tasks, including mathematical reasoning, code generation, and commonsense reasoning. We also conduct detailed ablation studies to assess the contribution of each core component in our framework. Owing to space limitations, additional details and results are included in the supplementary materials.

### 4.1 EXPERIMENT SETTINGS

**Datasets.** We evaluate *FedMomentum* across ten tasks, spanning three domains:

1. Math Reasoning. We employ MetaMathQA dataset (Yu et al., 2024) to fine-tune the base model for math reasoning, and evaluate on the test sets of GSM-8K (Cobbe et al., 2021) and MATH (Hendrycks et al., 2021).

2. Commonsense Reasoning. We fine-tune the base model on Commonsense170K dataset (Hu et al., 2023) and evaluate on 8 commonsense datasets, including BoolQ (Clark et al., 2019), PIQA (Bisk et al., 2020), SIQA (Sap et al., 2019), HellaSwag (Zellers et al., 2019), WinoGrande (Sakaguchi et al., 2020), ARC-e, ARC-c (Clark et al., 2018), and OBQA (Mihaylov et al., 2018).

3. Code Generation. We employ Code-Feedback dataset (Zheng et al., 2024) to fine-tune the base model for code generation, and evaluate by using the HumanEval (Chen et al., 2021) and MBPP (Austin et al., 2021) datasets.

For each task, we sample 10 local datasets at random following the non-IID setting as (Zhang et al., 2024), which uses Dirichlet distribution sampling ($D_k \sim Dir(\beta)$) and $\beta = 0.5$.

**Baseline.** We compare *FedMomentum* with vanilla FedAvg (FedIT) (Zhang et al., 2024) and three state-of-the-art baselines: FLoRA (Wang et al., 2024), FFA-LoRA (Sun et al., 2024), and FR-LoRA (Yan et al., 2025), to demonstrate its effectiveness and efficiency.

**Setup.** We utilize LLaMA2-7B (Touvron et al., 2023) for all tasks. Following (Li et al., 2025; Wang et al., 2025), we set the LoRA rank $r = 32$ and the scaling factor $\alpha = 64$. We use the default AdamW optimizer with a learning rate $lr = 3 \times 10^{-4}$.

We conduct federated fine-tuning with 10 clients. In each communication round, every client performs a local training consisting of 10 steps with a batch size of $b = 16$. To account for the varying fine-tuning difficulty across tasks, as reflected by different convergence speeds of training loss, we adopt task-specific communication round settings. Specifically, we run 50 rounds for math reasoning, 20 rounds for commonsense reasoning, and 30 rounds for code generation, ensuring sufficient convergence of training loss for each task.

## 4.2 EXPERIMENT RESULTS

**Math Reasoning Task.** As shown in Table 2, our proposed method *FedMomentum* achieves the best performance across all metrics in the math reasoning task, exceeding existing baselines in both the average score and individual accuracy. Particularly, in GSM8K, *FedMomentum* achieves 34.22% accuracy, representing a relative improvement of 18.0% over the second-best method FLoRA (29.06%), and a remarkable 219.3% improvement over FedIT (10.72%). Notably, FedIT performs even worse than the pre-trained model on GSM8K. This is due to the non-IID nature of client data, which amplifies aggregation noise under its separate $A/B$ aggregation scheme. In contrast, *FedMomentum* aggregates $B_i A_i$ without noise and uses SVD to reconstruct new LoRA modules, which

Table 2: Experiment results on math reasoning. The highest accuracy for each task is highlighted in bold, and the second-best result is underlined. Avg indicates the average result of corresponding metrics.

| Method | GSM8K | MATH | Avg. |
|---|---|---|---|
| Pre-trained | $13.87 \pm 0.95$ | $2.50 \pm 0.22$ | 8.19 |
| FedIT | $10.72 \pm 0.85$ | $3.84 \pm 0.27$ | 8.75 |
| FLoRA | $29.06 \pm 1.25$ | $3.86 \pm 0.27$ | 16.53 |
| FFA-LoRA | $25.40 \pm 1.20$ | $4.18 \pm 0.28$ | 14.91 |
| FRLoRA | $26.18 \pm 1.14$ | $3.97 \pm 0.26$ | 9.15 |
| *FedMomentum* | $\mathbf{34.22 \pm 1.31}$ | $\mathbf{4.44 \pm 0.29}$ | 19.99 |

preserves training momentum and contributes to its superior convergence and accuracy. Figure 4 shows that *FedMomentum* demonstrates an early and pronounced advantage in convergence speed, rapidly outperforming all baselines in training loss and maintaining this lead consistently across communication rounds.

To better understand this convergence behavior, we analyze the singular value spectrum and the number of major and residual components obtained by decomposing the aggregated updates in *FedMomentum*. Figure 5 shows that, as training progresses, the number of major components converges to $r$ and the number of residual components decreases to 0, suggesting that client updates are increasingly aligned within a lower-dimensional subspace. The moderate gap among the major singular values ensures that no direction dominates the low-rank reconstruction, helping to prevent energy collapse or imbalance in the reconstructed LoRA modules. As training progresses, this gap further

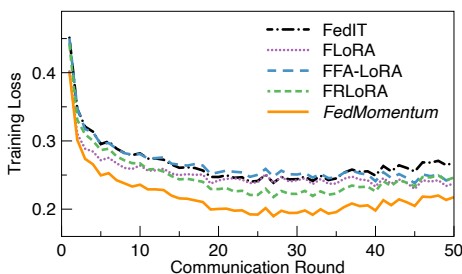

Figure 4: Training loss for math reasoning task.

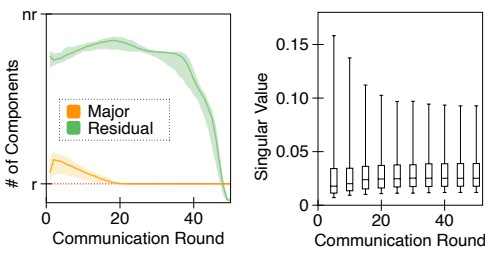

Figure 5: Statistical analysis for the aggregated updates of *FedMomentum*.

Table 3: Experiment results on eight commonsense reasoning tasks.

| Method | BoolQ | PIQA | SIQA | HellaSwag | WinoGrande | ARC-e | ARC-c | OBQA | Avg. |
|---|---|---|---|---|---|---|---|---|---|
| Pre-trained | 80.34 | **79.54** | 50.72 | **78.37** | 73.72 | 79.76 | 49.32 | 45.20 | 67.12 |
| FedIT | 82.10 | 78.35 | 53.23 | 76.63 | 74.30 | 80.52 | 50.64 | 47.70 | 67.93 |
| FLoRA | 76.53 | 77.15 | 48.72 | 73.54 | 70.72 | 76.64 | 43.43 | 42.10 | 63.60 |
| FFA-LoRA | **82.88** | 78.67 | 51.85 | 76.30 | 74.27 | 81.00 | 49.79 | 45.20 | 67.49 |
| FRLoRA | 80.02 | 77.40 | 51.90 | 72.30 | 72.57 | 79.55 | 47.99 | 47.90 | 66.20 |
| *FedMomentum* | 82.77 | 79.03 | **54.48** | 77.54 | **75.81** | 82.16 | 51.37 | 49.00 | **69.02** |

narrows, suggesting increasingly balanced update magnitudes along the principal directions. These trends collectively indicate that client updates are aligning within a compact and stable subspace, facilitating convergence toward a shared global model.

**Commonsense Reasoning Task.** To further validate the generalization ability of *FedMomentum*, we conduct experiments on more challenging commonsense reasoning tasks, evaluated across eight standard benchmarks. As shown in Table 3, *FedMomentum* achieves the highest accuracy on 5 out of 8 datasets and ranks second on the remaining 3, with only marginal gaps from the top-performing methods. Moreover, *FedMomentum* achieves the highest average accuracy of 69.02%, outperforming the best baseline (FedIT, 67.93%) by 1.09 points, further demonstrating the robustness and generalizability of our approach across diverse commonsense reasoning benchmarks.

**Code Generation Task.** Table 4 shows that our method *FedMomentum* achieves the highest score on both HumanEval (17.07%) and MBPP (25.60%), which leads to a 4.96% relative improvement over the second-best method in average code generation accuracy. FedIT and FLoRA struggle on both datasets with low accuracy. FRLoRA and FFA-LoRA show competitive results on one task but fail to generalize well across tasks. In contrast, *FedMomentum* retains the structural expressiveness of LoRA, resulting in better model quality.

## 4.3 ABLATION STUDY

To validate the effectiveness of key components in our method, we conduct ablation studies on the MetaMathQA benchmark with LLaMA2-7B and report the results in Table 5. Specifically, we remove the balanced allocation of singular values (denoted as *w/o* balance) and the residual term (denoted as *w/o* residual) to isolate their individual contributions.

Without balancing the singular values between $A$ and $B$ (*i.e.*, using unbalanced reconstruction $B = U_r \Sigma_r$, $A = V_r^\top$), the average accuracy is reduced from 19.99% to 12.63%, with a notable degradation of 12.61 percentage points on GSM8K. This validates our hypothesis that gradient imbalance arising from the skewed singular spectrum adversely affects convergence. The balanced decomposition (*i.e.*, distributing $\sqrt{\Sigma}$ to both $A$ and $B$) mitigates this issue, ensuring better gradient flow and training stability during local updates.

Removing the residual component leads to a performance drop, reducing the average accuracy from 19.99% to 18.02%. The observed performance drop confirms that the residual term, though diminishing over training (as shown in Figure 5), captures update directions not recoverable by fixed-rank

Table 4: Experiment results on code generation.

| Method | HumanEval | MBPP | Avg. |
|---|---|---|---|
| Pre-trained | 12.19 | 23.40 | 17.80 |
| FedIT | 15.24 | 24.00 | 19.62 |
| FLoRA | 13.41 | 23.00 | 18.21 |
| FFA-LoRA | 15.85 | 24.80 | 20.33 |
| FRLoRA | 16.46 | 24.00 | 20.23 |
| *FedMomentum* | **17.07** | **25.60** | **21.34** |

Table 5: Experiment results of ablation study on math reasoning tasks.

| Method | GSM8K | MATH | Avg. |
|---|---|---|---|
| Pre-trained | $13.87 \pm 0.95$ | $2.50 \pm 0.22$ | 8.19 |
| *FedMomentum* | $\mathbf{34.22 \pm 1.31}$ | $\mathbf{4.44 \pm 0.29}$ | **19.99** |
| *w/o* balance | $21.61 \pm 1.11$ | $3.64 \pm 0.29$ | 12.63 |
| *w/o* residual | $32.53 \pm 1.29$ | $3.50 \pm 0.28$ | 18.02 |

approximation alone. It thus provides a complementary signal that enhances the expressiveness of global updates, especially in the early stages when client updates span a higher-dimensional space.

# 5 RELATED WORK

Parameter-efficient fine-tuning (PEFT) techniques such as Low-Rank Adaptation (LoRA) have been adapted to federated learning (FL), aiming to reduce communication and computational costs. FedIT (Zhang et al., 2024) aggregates LoRA matrices $A$ and $B$ separately, but such independent averaging introduces aggregation noise, deviating from the true LoRA objective. FLoRA (Wang et al., 2024) addresses this by merging the local low-rank updates directly into the backbone model, achieving noise-free aggregation. However, its round-wise reinitialization of LoRA modules discards learned structures and slows convergence. FFA-LoRA (Sun et al., 2024) freezes the $A$ matrix and only updates and aggregates $B$, reducing noise but severely limiting the expressive capacity of local updates. FRLoRA (Yan et al., 2025) adds a residual term to correct aggregation noise, but the new LoRA modules may still diverge from the original updating direction, weakening training dynamics. In contrast, our method retains both structural fidelity and training momentum by leveraging an SVD-based aggregation mechanism that reconstructs LoRA matrices from aggregated updates.

**Computation-Efficient Federated Tuning.** Another line of work focuses on improving the computational efficiency of federated fine-tuning. FwdLLM (Xu et al., 2024) proposes a forward-only tuning paradigm that bypasses backpropagation and saves both memory and computation. FedPFT (Peng et al., 2024) introduces progressive fine-tuning with early exit strategies to accelerate convergence. FedBiOT (Wu et al., 2024) exploits bidirectional knowledge transfer and lightweight optimization to reduce client burden. While these approaches improve efficiency, they do not specifically address the structural preservation of LoRA updates or aggregation consistency.

**Resource-Aware Federated Adaptation.** To cope with device heterogeneity in FL, recent methods propose adaptive or flexible LoRA mechanisms. FlexLoRA (Bai et al., 2024) dynamically adjusts the LoRA rank per client, enabling fair and efficient training across heterogeneous devices. HETLoRA (Cho et al., 2024) supports different LoRA structures across clients while coordinating training via weighted aggregation. AFLoRA (Zhou et al., 2025) uses an adaptive LoRA controller to select appropriate modules based on local resource budgets. FedRA (Su et al., 2024) incorporates runtime profiling and resource-awareness into the aggregation process. These methods primarily focus on heterogeneous system conditions, while our approach emphasizes aggregation correctness and training momentum under standard federated setups.

# 6 CONCLUSION

In this paper, we identify a previously overlooked challenge in federated fine-tuning: the loss of training momentum due to aggregation noise and loss of structural expressiveness of LoRA. This phenomenon severely hinders the convergence and model performance in federated settings. We propose *FedMomentum*, a novel SVD-based aggregation algorithm that enables momentum-aware LoRA updating. By preserving the global update direction through low-rank matrix aggregation and decomposition, our method effectively mitigates the momentum loss issue. Extensive experiments on diverse tasks and models demonstrate that *FedMomentum* consistently achieves faster convergence and better accuracy compared to existing baselines.

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

## A  ADDITIONAL EXPERIMENTS AND SETUP DETAILS

### A.1  DATASETS, METRICS, AND ENVIRONMENTS

#### A.1.1  DATASET DETAILS.

We fine-tune the model on a set of publicly available instruction datasets covering mathematical reasoning, commonsense reasoning, and code generation tasks. The details are as follows:

- **MetaMathQA** (Yu et al., 2024): A large-scale math reasoning dataset of approximately 395k problems, constructed by augmenting the training datasets of GSM8K (Cobbe et al., 2021) and MATH (Hendrycks et al., 2021). It thus serves as a diverse and high-quality supervision dataset, specifically tailored for fine-tuning LLMs on mathematical reasoning tasks.

- **Commonsense170K** (Hu et al., 2023): A commonsense reasoning dataset containing 170k true/false questions. It covers diverse types of commonsense knowledge, including physical reasoning, social interactions, intentions, causes and effects, and everyday event understanding.

- **Code-Feedback** (Zheng et al., 2024): A code-centric dialogue dataset containing 66k multi-turn interactions between users and assistants. It covers diverse coding tasks such as writing, debugging, and refactoring code, and is designed to facilitate instruction tuning and feedback-driven refinement for code generation models.

To comprehensively assess model performance across different domains, we consider benchmarks from three categories: math reasoning, commonsense reasoning, and code generation. Details of each dataset are provided below:

- **GSM8K** (Cobbe et al., 2021): A benchmark for evaluating multi-step arithmetic reasoning, consisting of grade-school math word problems with solutions expressed in natural language.

- **MATH** (Hendrycks et al., 2021): A challenging benchmark for evaluating advanced mathematical reasoning, consisting of high-school competition problems across diverse domains such as algebra, geometry, and calculus.

- **BoolQ** (Clark et al., 2019): A binary question answering dataset derived from naturally occurring yes/no questions and Wikipedia passages.

- **PIQA** (Bisk et al., 2020): A benchmark for physical commonsense reasoning that evaluates a model's ability to choose the more plausible solution to everyday tasks. It focuses on physical interactions that challenge the understanding of intuitive, material-based problem solving.

- **SIQA** (Sap et al., 2019): A multiple-choice benchmark for social commonsense reasoning, where models are required to infer the most appropriate response to questions about people's actions and their social motivations or implications.

- **HellaSwag** (Zellers et al., 2019): A challenging multiple-choice benchmark for sentence completion, designed to evaluate grounded commonsense inference. The task requires a nuanced understanding of context and everyday knowledge.

- **WinoGrande** (Sakaguchi et al., 2020): A binary-choice benchmark for commonsense-based pronoun resolution, designed to improve scale and robustness over the original Winograd Schema Challenge (Levesque et al., 2012).

- **ARC-e and ARC-c** (Clark et al., 2018): Benchmarks for scientific question answering in a multiple-choice format. ARC-e evaluates basic science knowledge and reasoning at the elementary level, while ARC-c contains more difficult questions requiring advanced commonsense and deductive reasoning.
- **OBQA** (Mihaylov et al., 2018): A multiple-choice question answering benchmark that evaluates a model's ability to integrate elementary-level science knowledge with open-domain commonsense reasoning.
- **HumanEval** (Chen et al., 2021): A code generation benchmark evaluating whether models can produce Python functions that pass predefined test cases.
- **MBPP** (Austin et al., 2021): A code generation dataset consisting of simple Python programming tasks with corresponding unit tests for evaluating correctness.

### A.1.2 METRICS.

We report *accuracy* for all multiple-choice and QA datasets, including GSM8K, MATH, BoolQ, PIQA, SIQA, HellaSwag, WinoGrande, ARC, and OBQA. For code generation benchmarks (HumanEval and MBPP), we use *pass@1* as the evaluation metric.

### A.1.3 COMPUTER RESOURCES.

All experiments are implemented using PyTorch and conducted on an Ascend 910B GPU with 64 GB of memory, running on a HiSilicon Kunpeng-920 CPU with Linux v5.10.0.

## A.2 OTHER DETAILS

### A.2.1 EVALUATION SETTINGS

We repeat each experiment three times and show the averaged results. For evaluation, we employ two widely used task-specific frameworks: lm-evaluation-harness[1] for math reasoning and commonsense reasoning tasks, and bigcode-evaluation-harness[2] for code generation tasks.

### A.2.2 RESIDUAL-ENERGY THRESHOLD

As shown in Figure 2, the tail singular values are over two orders of magnitude smaller than the leading ones. Their squared contribution is therefore $< 0.01\%$ of the total energy, meaning that the top components already capture essentially all useful information in the aggregated update.

Also, considering that SVD-based methods typically regard $> 99\%$ cumulative energy as sufficient to preserve the matrix structure, we adopt a conservative threshold of $99.99\%$ to avoid introducing any approximation bias during aggregation. This ensures that all meaningful directions are retained, while only removing numerically negligible components.

Formally, we compute $E(t) = \frac{\sum_{j=1}^{t} \sigma_j^2}{\sum_{j=1}^{nr} \sigma_j^2}$, and define $r_{\text{eff}} = \min\{t : E(t) \geq \tau\}$, $s = r_{\text{eff}} - r$, and $\tau = 0.9999$.

Although a smaller threshold would further reduce communication, our priority is avoiding bias and ensuring stable training, and $99.99\%$ achieves this while keeping the residual rank very small in practice.

## A.3 SUPPLEMENTARY EXPERIMENT RESULTS

### A.3.1 COMPUTATIONAL OVERHEAD.

To demonstrate the efficiency of *FedMomentum*, we conduct experiments on a math reasoning task under default settings. Table 6 presents the average per-round runtime of LoRA aggregation and local model updating across different methods. Compared to prior approaches, our proposed method

---

[1] https://github.com/EleutherAI/lm-evaluation-harness
[2] https://github.com/bigcode-project/bigcode-evaluation-harness

Table 6: Average runtime (in seconds) of different methods.

| Method | Aggregation Time (s) | Updating Time (s) |
|---|---|---|
| FedIT | 0.10 | 0.02 |
| FLoRA | 0.05 | 0.03 |
| FFA-LoRA | 0.06 | 0.02 |
| FRLoRA | 0.15 | 0.03 |
| *FedMomentum* | 0.60 | 0.03 |
| *w/* exact SVD | >1000 | 0.03 |

Table 7: Results of ablation study on commonsense reasoning tasks.

| Method | BoolQ | PIQA | SIQA | HellaSwag | WinoGrande | ARC-e | ARC-c | OBQA | Avg. |
|---|---|---|---|---|---|---|---|---|---|
| Pre-trained | 80.34 | **79.54** | 50.72 | **78.37** | 73.72 | 79.76 | 49.32 | 45.20 | 67.12 |
| *FedMomentum* | 82.77 | 79.03 | **54.48** | 77.54 | **75.81** | **82.16** | **51.37** | **49.00** | 69.02 |
| *w/o* balance | 82.13 | 78.56 | 53.15 | 76.54 | 73.88 | 80.37 | 49.74 | 46.90 | 67.67 |
| *w/o* residual | 81.71 | 77.91 | 52.64 | 76.28 | 74.78 | 80.93 | 49.36 | 48.40 | 67.75 |

*FedMomentum* achieves a competitive aggregation time (0.60s) while maintaining a low local updating cost (0.03s). Although the aggregation time is marginally higher than baseline methods such as FedIT and FLoRA, the difference is negligible in practice. Meanwhile, *FedMomentum* demonstrates significantly faster convergence and better final performance, as shown in the main body of the paper.

To further assess the practicality of our design, we also report the runtime when replacing our randomized SVD with exact SVD computation. The aggregation time in this setting exceeds 1000 seconds per round, rendering it infeasible for real-world deployments. These results highlight the efficiency advantage of randomized SVD over exact SVD, reducing aggregation time by several orders of magnitude while maintaining effective LoRA aggregation.

### A.3.2 Additional Ablation Study Results.

To further assess the generality of our design choices, we conduct additional ablation studies on the commonsense reasoning and the code generation tasks and report the results.

On the commonsense reasoning, Table 7 shows that removing the balanced allocation of singular values (*w/o* balance) leads to an average performance drop from 69.02% to 67.67%, while removing the residual component (*w/o* residual) results in a similar reduction to 67.75%.

On the code generation, Table 8 shows that ablating the balanced allocation of singular values (*w/o* balance) reduces the average *pass@*1 accuracy from 21.34% to 19.22%, while removing the residual term (*w/o* residual) leads to a smaller decline to 20.63%. These results suggest that both components contribute to stable improvements across diverse tasks.

### A.3.3 Impact of Different Ranks.

To evaluate the generalization of different methods under varying LoRA capacities, we compare their performance on the math reasoning task at ranks 16, 32, and 64. We use MetaMathQA for fine-tuning and GSM8K for evaluation. Figure 6 shows that *FedMomentum* consistently outperforms all baselines across different ranks, demonstrating strong robustness to the choice of LoRA rank. In particular, it maintains high accuracy even under low-rank settings (*e.g.*, 32.75% at rank 16), and continues to improve with larger ranks, showing its scalability and effectiveness.

To evaluate extremely low-rank settings, we also conduct additional experiments on the math reasoning task at ranks 1 and 2. As Figure 6 shown, all methods exhibit significantly degraded performance under such extremely constrained ranks (compared to rank 16/32/64), indicating that these ranks severely limit the representational capacity needed for reasoning tasks.

Table 8: Results of ablation study on code generation.

| Method | Code-Feedback | | |
|--------|-----------|------|------|
| | HumanEval | MBPP | Avg. |
| Pre-trained | 12.19 | 23.40 | 17.80 |
| *FedMomentum* | **17.07** | **25.60** | **21.34** |
| *w/o* balance | 14.63 | 23.80 | 19.22 |
| *w/o* residual | 16.46 | 24.80 | 20.63 |

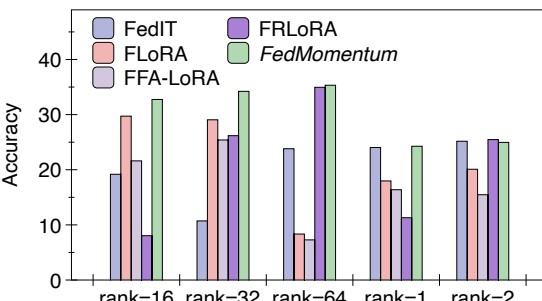

Figure 6: Comparison of different methods under different LoRA ranks on the math reasoning task.

Nevertheless, *FedMomentum* remains remarkably stable in this regime: With rank-1, *FedMomentum* achieves the best performance among all methods; with rank-2, *FedMomentum* remains close to the top-performing baseline.

These results show that although extremely low ranks reduce the overall effectiveness of federated fine-tuning, *FedMomentum* degrades more gracefully and maintains a competitive advantage even in this challenging setting.

### A.4 THE USE OF LARGE LANGUAGE MODELS

During the preparation of this paper, we used a large language model (ChatGPT) solely as a writing assistant for polishing grammar and improving the readability of the text. The model was not involved in research ideation, experimental design, implementation, analysis, or drawing conclusions. All scientific content and claims are the authors' own responsibility.

## B COMMUNICATION OVERHEAD ANALYSIS

Let $p_{\text{LoRA}} = \sum_{\ell \in \mathcal{L}} r(d_\ell + k_\ell)$ denote the total number of LoRA parameters across all instrumented layers $l$ (with per-layer shapes $W_\ell \in \mathbb{R}^{d_\ell \times k_\ell}$ and rank $r$), and let $p_{\text{full}} = \sum_{\ell \in \mathcal{L}} d_\ell k_\ell$ denote the total backbone parameter size. We also denote the number of clients by $n$.

Under this notation, the per-round, per-client communication costs (uplink + downlink) of the baselines can be summarized as follows:

**FedIT:** Each client uploads its local LoRA parameters $(A_\ell, B_\ell)$, and receives the aggregated global LoRA. This gives $p_{\text{FedIT}} = \underbrace{p_{\text{LoRA}}}_{\text{uplink}} + \underbrace{p_{\text{LoRA}}}_{\text{downlink}} \Rightarrow (1+1)p_{\text{LoRA}}$.

**FLoRA:** Each client still uploads its own LoRA parameters, but the server stacks all clients' low-rank modules to form an effective rank-$nr$ adapter and broadcasts the stacked LoRA back to all clients. Therefore, the downlink to each client is n times larger than in FedIT, $p_{\text{FLoRA}} = \underbrace{p_{\text{LoRA}}}_{\text{uplink}} + \underbrace{n \cdot p_{\text{LoRA}}}_{\text{downlink}} = (1+n) \cdot p_{\text{LoRA}}$.

**FFA-LoRA:** FFA-LoRA freezes all $A_\ell$ and only communicates $B_\ell \in \mathbb{R}^{d_\ell \times r}$. In terms of order, this reduces the LoRA payload by roughly a factor of two when $\sum_\ell d_\ell \approx \sum_\ell k_\ell$, leading to $p_{\text{FFA-LoRA}} \approx \underbrace{\frac{1}{2} p_{\text{LoRA}}}_{\text{uplink}} + \underbrace{\frac{1}{2} p_{\text{LoRA}}}_{\text{downlink}} = p_{\text{LoRA}}$.

**FRLoRA:** FRLoRA performs FedAvg on the LoRA parameters (as in FedIT) and communicates a full-size residual per layer to correct aggregation noise. Thus, $p_{\text{FRLoRA}} = \underbrace{p_{\text{LoRA}}}_{\text{uplink}} + \underbrace{p_{\text{LoRA}}}_{\text{downlink (LoRA part)}} + \underbrace{p_{\text{full}}}_{\text{downlink (dense residual)}} = 2 \cdot p_{\text{LoRA}} + p_{\text{full}}$. where the additional $p_{\text{full}}$ term dominates for large LLMs, making FRLoRA no longer communication-efficient.

***FedMomentum***: Each client uploads its local LoRA parameters once per round (same as FedIT), *i.e.*, $p_{\text{LoRA}}$. On the downlink, the server sends back a low-rank reconstruction consisting of the major $r$ components (forming the new LoRA) and an additional residual subspace of rank $s$ (still in low-rank form, not full dense). This corresponds to an effective rank $r + s$ adapter, so the downlink size is $\frac{r+s}{r} \cdot p_{\text{LoRA}}$.

For comparison with FLoRA's rank-$nr$ stacking, it is convenient to write $\lambda = \frac{r+s}{nr}$. Since the total rank after aggregation is upper-bounded by $nr$, we have $0 \leq s \leq (n-1)r$, which implies $\lambda = \frac{r+s}{nr} \in \left[\frac{1}{n}, 1\right]$.

Then, the per-round, per-client communication cost of *FedMomentum* can be written as $p_{FedMomentum} = \underbrace{p_{\text{LoRA}}}_{\text{uplink}} + \underbrace{\lambda n \cdot p_{\text{LoRA}}}_{\text{downlink}} = (1 + \lambda n) \cdot p_{\text{LoRA}}$. In other words:

- When $s = 0$ (no residual subspace), $\lambda = 1/n$ and *FedMomentum* degenerates to the FedIT communication level: $(1 + \lambda n) \cdot p_{\text{LoRA}} = (1 + 1) \cdot p_{\text{LoRA}}$;
- When $s = (n-1)r$ (maximal residual), $\lambda = 1$ and the downlink matches FLoRA's stacked rank-$nr$ adapter: $(1 + \lambda n) \cdot p_{\text{LoRA}} = (1 + n) \cdot p_{\text{LoRA}}$. Therefore, ours achieves lower communication cost than FLoRA while delivering consistently better convergence, and in some cases its overhead is comparable to FedIT, making it a more communication-efficient and effective approach for federated LoRA fine-tuning.

## C  PRIVACY DISCUSSION

Our method does not introduce additional privacy leakage compared with existing baselines. *FedMomentum* only applies SVD to the aggregated update $\Delta W = \sum_{i=1}^{n} B_i A_i$, which is the similar quantity that all baselines (FedIT, FLoRA, and FRLoRA) already share or broadcast. No client-specific update is ever transmitted.

SVD does not reveal more information than the matrix itself. Any party receiving the aggregated update in baseline methods can already run SVD locally. Thus, sending the SVD factors (or residuals derived from them) does not expose more information than existing approaches. Residual components are also derived from the same aggregated update and contain no per-client identifiable structure.

Therefore, *FedMomentum* maintains the same privacy level as existing federated LoRA methods and does not increase the risk of client information leakage.

