# OpenReview forum: "FedMomentum: Preserving LoRA Training Momentum in Federated Fine-Tuning"
_ICLR.cc/2026/Conference — Submitted to ICLR 2026_

### Official Review · Reviewer_hXhJ · 2025-10-18

**Soundness:** 2
**Presentation:** 2
**Contribution:** 1
**Rating:** 4
**Confidence:** 5

**Summary:**

FedMomentum is a federated fine-tuning framework for LoRA-based models that aggregates client updates in the multiplication of LoRA-B and LoRA-A space and then applies SVD to the aggregated updates. The server divides singular components to major components, residual components, and negligible components, and reconstructs LoRA using major components. Residual components are tracked as a residual term and can be merged into the backbone. Experiments across multiple LLM tasks (e.g., reasoning and code generation) emphasize convergence speed and final accuracy.

**Strengths:**

1. The experiments show consistent gains across multiple datasets.

**Weaknesses:**

1. FlexLoRA[1], which also uses SVD-based aggregation, though mentioned in the related works, should be thoroughly discussed, compared and included as an important baseline. The main incremental contribution of FedMomentum is its insights, but the method is highly similar to the closest work FlexLoRA and has not been fully compared; in the absence of rigorous positioning and systematic ablation, the existing performance improvement is difficult to be clearly attributed.
2. It is not clear how to determine the number of residual components, could the author formally define it?
3. SVD is not new in aggregation for LoRA in FL. The computational and communication overhead induced by SVD should be studied comprehensively in different FL scenarios (different models, different client numbers).
4. Tables 2–5 should also report, for each method, the communication cost required to reach the corresponding accuracy.

[1] Federated Fine-tuning of Large Language Models under Heterogeneous Tasks and Client Resources. Bai et al. NeurIPS 2024

**Questions:**

1. Could the author evaluate the impact of extreme low rank (rank-1 and rank-2)?
2. Could the author scale up the client number to 30 or more to test the effectiveness of the proposed aggregation?
3. How is the non-IID setting constructed for datasets without labels, such as GSM8k or HumanEval?

---

> ### Author Response · Authors · 2025-11-24
> **Author response to Reviewer hXhJ (Part 1 of 3)**
>
> > Q1: FlexLoRA should be thoroughly discussed, compared and included as an important baseline. SVD is not new in aggregation for LoRA in FL.
>
> We thank the reviewer for highlighting the relation to FlexLoRA and the use of SVD in prior LoRA-based methods. To avoid misunderstanding, we clarify our positioning and contributions as follows:
>
> **(1) Our contribution is not SVD itself, but the discovery and resolution of momentum loss in federated LoRA fine-tuning.**
>
> While SVD has indeed been used in LoRA initialization and in FlexLoRA for adapting heterogeneous client ranks, *none* of these methods study the structural inconsistency caused by averaging LoRA updates in FL, nor identify that this mismatch leads to loss of training momentum and degraded convergence.
>
> Our key contribution is the first to: 1) identify the structural destruction → momentum loss phenomenon in federated LoRA aggregation, 2) explain its mechanism, and 3) propose a structure-preserving, momentum-preserving aggregation rule.
>
> SVD is only the tool we use to reconstruct aligned low-rank directions; the insight and method lie in preserving LoRA structure and momentum, not in applying SVD.
>
> **(2) FlexLoRA addresses a different problem and is not directly comparable.**
>
> FlexLoRA uses SVD to map client updates of different ranks into a unified rank space.
>
> In contrast, FedMomentum assumes a fixed rank and focuses on preserving the correct BA structure and continual optimization direction across rounds. These address orthogonal challenges. FlexLoRA is not designed to tackle structural misalignment or momentum continuity, which is central to our method.
>
> This conceptual difference explains why the two methods are not interchangeable.
>
> **(3) On SVD overhead.**
>
> We only apply randomized SVD on the aggregated update once per round. As shown in our experiments and the following communication cost analysis, the computational and communication cost are modest and comparable to existing baselines. Moreover, in practice, the number of the local model updates used for aggregation will not be large since there is always partial participation when the total number of clients is large.
>
> We will revise the paper to clarify these conceptual distinctions.
>
>
>
> > Q2: It is not clear how to determine the number of residual components
>
> As shown in Figure 2, the tail singular values are over two orders of magnitude smaller than the leading ones. Their squared contribution is therefore <0.01% of the total energy, meaning that the top components already capture essentially all useful information in the aggregated update.
>
> Also, considering that SVD-based methods typically regard >99% cumulative energy as sufficient to preserve the matrix structure, we adopt a conservative threshold of 99.99% to avoid introducing any approximation bias during aggregation. This ensures that all meaningful directions are retained, while only removing numerically negligible components.
>
> Formally, we compute $E(t)=\frac{\sum\_{j=1}^t\sigma\_j^2}{\sum\_{j=1}^{nr}\sigma\_j^2},$ and define $r\_{\text{eff}}=\min\{t:E(t)\ge\tau\},\quad s=r\_{\text{eff}}-r,\quad \tau=0.9999.$
>
> Although a smaller threshold would further reduce communication, our priority is avoiding bias and ensuring stable training, and 99.99% achieves this while keeping the residual rank very small in practice.

---

> ### Author Response · Authors · 2025-11-24
> **Author response to Reviewer hXhJ (Part 2 of 3)**
>
> > Q3: Tables 2–5 should also report, for each method, the communication cost required to reach the corresponding accuracy.
>
> We thank the reviewer for pointing out the importance of communication analysis. We now provide a unified comparison in terms of per-round, per-client communication volume.
>
> Let $p\_{\text{LoRA}} = \sum\_{\ell\in\mathcal{L}} r(d\_\ell + k\_\ell)$ denote the total number of LoRA parameters across all instrumented layers $l$ (with per-layer shapes $W\_\ell\in\mathbb{R}^{d\_\ell\times k\_\ell}$ and rank $r$), and let $p\_{\text{full}} = \sum\_{\ell\in\mathcal{L}} d\_\ell k\_\ell$ denote the total backbone parameter size. We also denote the number of clients by $n$.
>
> Under this notation, the per-round, per-client communication costs (uplink + downlink) of the baselines can be summarized as follows:
>
> **FedIT:** Each client uploads its local LoRA parameters ($A\_\ell,B\_\ell$), and receives the aggregated global LoRA. This gives $p\_{\text{FedIT}}=\underbrace{p\_{\text{LoRA}}}\_{\text{uplink}}+\underbrace{p\_{\text{LoRA}}}\_{\text{downlink}}\Rightarrow(1+1)p\_{\text{LoRA}}.$
>
> **FLoRA:** Each client still uploads its own LoRA parameters, but the server stacks all clients' low-rank modules to form an effective rank-$nr$ adapter and broadcasts the stacked LoRA back to all clients. Therefore, the downlink to each client is n times larger than in FedIT, $p\_{\text{FLoRA}}
> =\underbrace{p\_{\text{LoRA}}}\_{\text{uplink}}+\underbrace{n\cdot p\_{\text{LoRA}}}\_{\text{downlink}}
> = (1+n)\cdot p\_{\text{LoRA}}.$
>
> FFA-LoRA: FFA-LoRA freezes all $A\_\ell$ and only communicates $B\_\ell\in\mathbb{R}^{d\_\ell\times r}$. In terms of order, this reduces the LoRA payload by roughly a factor of two when $\sum\_\ell d\_\ell \approx \sum\_\ell k\_\ell$, leading to $p\_{\text{FFA-LoRA}}\approx\underbrace{\tfrac{1}{2}p\_{\text{LoRA}}}\_{\text{uplink}}+\underbrace{\tfrac{1}{2}p\_{\text{LoRA}}}\_{\text{downlink}}= p\_{\text{LoRA}}.$
>
>  FRLoRA: FRLoRA performs FedAvg on the LoRA parameters (as in FedIT) and communicates a full-size residual per layer to correct aggregation noise. Thus, $p\_{\text{FRLoRA}}=\underbrace{p\_{\text{LoRA}}}\_{\text{uplink}}+\underbrace{p\_{\text{LoRA}}}\_{\text{downlink (LoRA part)}}+\underbrace{p\_{\text{full}}}\_{\text{downlink (dense residual)}}= 2\cdot p\_{\text{LoRA}} + p\_{\text{full}}.$ where the additional $p\_{\text{full}}$ term dominates for large LLMs, making FRLoRA no longer communication-efficient.
>
> For FedMomentum, each client uploads its local LoRA parameters once per round (same as FedIT), i.e., $p\_{\text{LoRA}}$. On the downlink, the server sends back a low-rank reconstruction consisting of the major $r$ components (forming the new LoRA) and an additional residual subspace of rank $s$ (still in low-rank form, not full dense). This corresponds to an effective rank $r+s$ adapter, so the downlink size is $\frac{r+s}{r}\cdot \cdot p\_{\text{LoRA}}$.
>
> For comparison with FLoRA's rank-$nr$ stacking, it is convenient to write $\lambda = \frac{r+s}{nr}$.
> Since the total rank after aggregation is upper-bounded by $nr$, we have $0 \le s \le (n-1)r$, which implies $\lambda = \frac{r+s}{nr} \in \left[\frac{1}{n},1\right].$
>
> Then, the per-round, per-client communication cost of FedMomentum can be written as
> $p\_{\text{FedMomentum}}=\underbrace{p\_{\text{LoRA}}}\_{\text{uplink}}+\underbrace{\lambda n\cdot p\_{\text{LoRA}}}\_{\text{downlink}}= (1+\lambda n)\cdot p\_{\text{LoRA}}.$ In other words:
>
> * when $s=0$ (no residual subspace), $\lambda = 1/n$ and FedMomentum degenerates to the FedIT communication level: $(1+\lambda n)\cdot p\_{\text{LoRA}} = (1+1)\cdot p\_{\text{LoRA}};$
>
> * when $s=(n-1)r$ (maximal residual), $\lambda=1$ and the downlink matches FLoRA's stacked rank-$nr$ adapter: $(1+\lambda n)\cdot p\_{\text{LoRA}} = (1+n)\cdot p\_{\text{LoRA}}.$
>   Therefore, ours achieves lower communication cost than FLoRA while delivering consistently better convergence, and in some cases its overhead is comparable to FedIT, making it a more communication-efficient and effective approach for federated LoRA fine-tuning.

---

> ### Author Response · Authors · 2025-11-24
> **Author response to Reviewer hXhJ (Part 3 of 3)**
>
> > Q4: Could the author evaluate the impact of extreme low rank (rank-1 and rank-2)?
>
> We appreciate the reviewer's suggestion to evaluate extremely low-rank settings. We have now conducted additional experiments on the MATH Reasoning task with rank = 1 and rank = 2. As expected, all federated LoRA methods exhibit significantly degraded performance under such extremely constrained ranks (compared to rank 16/32/64), indicating that these ranks severely limit the representational capacity needed for reasoning tasks.
>
> Nevertheless, FedMomentum remains remarkably stable in this regime: With rank-1, FedMomentum achieves the best performance among all methods; with rank-2, FedMomentum remains close to the top-performing baseline.
>
> These results show that although extremely low ranks reduce the overall effectiveness of federated fine-tuning, FedMomentum degrades more gracefully and maintains a competitive advantage even in this challenging setting. We will include the complete results in the supplementary material.
>
> | Method      | r=1   | r=2   |
> | ----------- | ----- | ----- |
> | FedIT       | 24.03 | 25.17 |
> | FLoRA       | 17.97 | 20.09 |
> | FFA-LoRA    | 16.38 | 15.47 |
> | FRLoRA      | 11.3  | 25.47 |
> | FedMomentum | 24.26 | 24.96 |
>
>
>
> > Q5: Could the author scale up the client number to 30 or more to test the effectiveness of the proposed aggregation?
>
> We thank the reviewer for the suggestion to evaluate larger client counts. In existing federated LoRA methods (e.g., FLoRA, and FRLoRA), the commonly adopted scale is 5-10 clients, and we follow this standard setting to ensure fair and directly comparable evaluation across methods.
>
> Scaling to 30+ clients substantially increases computational costs. Given our current resource constraints, we are unfortunately unable to include such large-scale experiments within the rebuttal period. We appreciate the reviewer's understanding and will consider expanding to higher client counts as future work.
>
> > Q6: How is the non-IID setting constructed for datasets without labels, such as GSM8k or HumanEval?
>
> Our non-IID data construction follows the standard practice used in prior federated NLP methods [FedIT, and FLoRA]. For datasets without labels (e.g., GSM8K, and HumanEval), we treat the entire dataset as a single pseudo-category. We then apply a Dirichlet($\alpha$)-based partition over sample indices: for each client i, a sampling proportion $p_i$ is drawn from a Dirichlet distribution with concentration parameter $\alpha$. This yields uneven sample allocations across clients, where smaller $\alpha$ induces more extreme non-IID splits. Although not label-based, this procedure preserves the standard FL evaluation practice for text datasets without explicit class annotations.

---

### Official Review · Reviewer_rTjD · 2025-10-30

**Soundness:** 3
**Presentation:** 3
**Contribution:** 3
**Rating:** 8
**Confidence:** 4

**Summary:**

The paper introduces FedMomentum, an SVD-based aggregation scheme for federated LoRA fine-tuning that (1) aggregates client updates in the correct BA form, (2) performs (randomized) SVD on the summed update to recover top-r principal components, (3) balances singular values across B and A to avoid gradient anisotropy, and (4) carries residual components that are later merged into the backbone to preserve semantics, thereby maintaining training momentum across rounds. Experiments on math, commonsense, and code benchmarks show faster convergence and higher accuracy than baselines.

**Strengths:**

1. Consistent improvements on math, commonsense, and code benchmarks.
2. Splitting Σ as Σ1/2 across B and A is a low-cost fix for singular-value skew.

**Weaknesses:**

1. All results are on LLaMA2-7B; behavior on newer or larger models is unknown.
2. Experiments use 10 clients with Dirichlet β=0.5. It’s unclear how momentum preservation holds under more extreme non-IID, partial participation, or straggler scenarios.

**Questions:**

1. How does FedMomentum perform on a recent larger model?
2. What residual-energy threshold and merge cadence are recommended across tasks?
3. Could residuals/SVD factors leak client characteristics?

---

> ### Author Response · Authors · 2025-11-24
> **Author response to Reviewer rTjD**
>
> > Q1: All results are on LLaMA2-7B; behavior on newer or larger models is unknown.
>
> We appreciate the reviewer's comment regarding the generalizability of our results to newer or larger models. It is important to note that the majority of existing methods in this area have primarily focused on validating methods using the RoBERTa model. Only a few studies have extended their evaluation to the scale of LLaMA-2-7B, which is already a relatively large model in current federated learning and fine-tuning research.
>
> In our work, we follow the existing standard in the field by performing experiments on LLaMA-2-7B to ensure consistency with prior research. Furthermore, based on our preliminary experiments, we have observed that models at the scale of LLaMA-2-7B, as well as even larger models like LLaMA-3-8B, already exhibit strong generalization performance. In fact, for a number of fine-tuning tasks outlined in the paper, we found that centralized fine-tuning and federated fine-tuning (with various strategies) did not result in significant performance improvements over the pre-trained model.
>
> This suggests that the performance of these larger models has already reached a point where further fine-tuning provides diminishing returns, particularly in the context of the tasks we evaluated. Therefore, we believe that our choice to focus on LLaMA-2-7B and to evaluate performance on such models is both consistent with existing work and sufficiently comprehensive for the purposes of this study.
>
> > Q2:  It's unclear how momentum preservation holds under more extreme non-IID, partial participation, or straggler scenarios.
>
> We thank the reviewer for raising this important point. Our current experiments use 10 clients with Dirichlet $\alpha=0.5$, following the same non-IID setting adopted in prior federated LoRA works such as FLoRA. We intentionally keep this setting to ensure fair and directly comparable results across baselines.
>
> Due to the substantial computational cost of federated fine-tuning on LLaMA-2-7B, we are unable to include additional experiments under more extreme non-IID, partial participation, or straggler conditions within the rebuttal period. We appreciate the reviewer's understanding.
>
> Importantly, our method is algorithmically compatible with these scenarios: the structure-preserving aggregation and momentum continuity mechanisms operate solely on the aggregated update and do not rely on full participation. Extending FedMomentum to partial participation and straggler-resilient settings is a natural next step, and we plan to explore these directions in future work.
>
>
>
> > Q3: What residual-energy threshold and merge cadence are recommended across tasks?
>
> **For the residual energy threshold:** As shown in Figure 2, the tail singular values are over two orders of magnitude smaller than the leading ones. Their squared contribution is therefore <0.01% of the total energy, meaning that the top components already capture essentially all useful information in the aggregated update.
>
> Also, considering that SVD-based methods typically regard >99% cumulative energy as sufficient to preserve the matrix structure, we adopt a conservative threshold of 99.99% to avoid introducing any approximation bias during aggregation. This ensures that all meaningful directions are retained, while only removing numerically negligible components.
>
> Formally, we compute $E(t)=\frac{\sum\_{j=1}^t\sigma\_j^2}{\sum\_{j=1}^{nr}\sigma\_j^2},$ and define $r\_{\text{eff}}=\min\{t:E(t)\ge\tau\},\quad s=r\_{\text{eff}}-r,\quad \tau=0.9999.$
>
> Although a smaller threshold would further reduce communication, our priority is avoiding bias and ensuring stable training, and 99.99% achieves this while keeping the residual rank very small in practice.
>
> **For merge cadence**, as described in Section 3.3, we merge the residual terms into the backbone each communication round.
>
>
>
> > Q4: Could residuals/SVD factors leak client characteristics?
>
> Our method does not introduce additional leakage compared with existing baselines.
>
> FedMomentum only applies SVD to the aggregated update $\Delta W = \sum_{i=1}^n B_i A_i,$ which is the similar quantity that all baselines (FedIT, FLoRA, and FRLoRA) already share or broadcast. No client-specific update is ever transmitted.
>
> SVD does not reveal more information than the matrix itself. Any party receiving the aggregated update in baseline methods can already run SVD locally. Therefore, sending the SVD factors (or residuals derived from them) does not expose more information than existing approaches.
>
> Residual components are also derived from the same aggregated update and contain no per-client identifiable structure.
>
> Thus, FedMomentum maintains the same privacy level as existing federated LoRA methods and does not increase the risk of client information leakage.

---

### Official Review · Reviewer_mVza · 2025-11-01

**Soundness:** 2
**Presentation:** 3
**Contribution:** 2
**Rating:** 4
**Confidence:** 4

**Summary:**

The paper proposes FedMomentum, a federated LoRA fine-tuning framework that aggregates delta weights (BA) across clients and then applies (randomized) SVD to (i) reconstruct LoRA modules from the top-r components with a balanced √Σ split between A and B to stabilize gradients, and (ii) carry forward residual components that are later merged into the backbone. This aims to avoid aggregation noise and preserve “training momentum” that is otherwise lost by reinitialization or partial freezing strategies. Experiments on math reasoning, commonsense reasoning, and code generation with LLaMA2-7B show faster convergence and higher accuracy than FedIT, FLoRA, FFA-LoRA, and FRLoRA; ablations indicate both the balanced split and residual path contribute to gains.

**Strengths:**

Clear problem framing around momentum loss in federated LoRA and a principled SVD-based fix.

Consistent improvements and faster convergence over strong baselines, with thorough ablations.

Practicality considered via randomized SVD and reporting of runtime overhead.

**Weaknesses:**

“Momentum” story isn’t operationalized:

The paper motivates momentum preservation with spectra/visuals but never measures optimization continuity directly (e.g., gradient-direction alignment across rounds, cosine to prior updates, curvature drift). Without such probes, the claimed mechanism remains a narrative rather than an evidenced cause.

Residuals add hidden communication/state costs:

FedMomentum keeps a residual subspace (until ~99% energy is retained) and ships it back for backbone merges. Early rounds can have sizable residual rank; yet the paper reports aggregation time, not the bytes/round for residuals vs. plain LoRA A/B. This makes “communication-efficient” hard to verify under realistic uplink constraints.

Fragile hyperparameters, little sensitivity analysis:

Key knobs (LoRA rank r, the residual energy threshold, randomized-SVD oversampling/power iterations, and the √Σ split) are fixed or lightly ablated. There’s no sweep showing accuracy vs. residual threshold, nor stability across seeds/clients—leaving robustness and reproducibility unclear.

**Questions:**

Please see weaknesses

---

> ### Author Response · Authors · 2025-11-24
> **Author response to Reviewer mVza (Part 1 of 3)**
>
> > Q1: "Momentum" story isn't operationalized. The claimed mechanism remains a narrative rather than an evidenced cause.
>
> We thank the reviewer for the suggestion. The paper already provides two complementary forms of evidence that operationalize momentum continuity in a way that is appropriate for LoRA-based FL:
>
> **(1) Behavioral evidence from centralized fine-tuning (Figure 1).**
>
> We show that breaking the LoRA structure (e.g., via repeated merge or reinitialization) leads to significantly slower convergence even in centralized training—precisely the expected symptom of optimization-direction discontinuity. In contrast, FedMomentum closely matches the centralized convergence curve, indicating preserved optimization continuity.
>
> **(2) Structural evidence inside FedMomentum (Figures 2 and 5).**
>
> Across rounds, FedMomentum maintains a stable top-r principal subspace and strongly dominant leading singular values, indicating that updates consistently align with a fixed low-rank direction. This stability is exactly the structural analogue of momentum preservation for LoRA updates, whose effective optimization directions lie in the BA subspace rather than in the full gradient space.
>
> Together, these behavioral and structural observations provide converging and operational evidence for momentum continuity in LoRA-based federated fine-tuning.

---

> ### Author Response · Authors · 2025-11-24
> **Author response to Reviewer mVza (Part 2 of 3)**
>
> > Q2: Residuals add hidden communication/state costs. FedMomentum is not “communication-efficient”
>
> We thank the reviewer for pointing out the importance of communication analysis. We now provide a unified comparison in terms of per-round, per-client communication volume.
>
> Let $p\_{\text{LoRA}} = \sum\_{\ell\in\mathcal{L}} r(d\_\ell + k\_\ell)$ denote the total number of LoRA parameters across all instrumented layers $l$ (with per-layer shapes $W\_\ell\in\mathbb{R}^{d\_\ell\times k\_\ell}$ and rank $r$), and let $p\_{\text{full}} = \sum\_{\ell\in\mathcal{L}} d\_\ell k\_\ell$ denote the total backbone parameter size. We also denote the number of clients by $n$.
>
> Under this notation, the per-round, per-client communication costs (uplink + downlink) of the baselines can be summarized as follows:
>
> **FedIT:** Each client uploads its local LoRA parameters ($A\_\ell,B\_\ell$), and receives the aggregated global LoRA. This gives $p\_{\text{FedIT}}=\underbrace{p\_{\text{LoRA}}}\_{\text{uplink}}+\underbrace{p\_{\text{LoRA}}}\_{\text{downlink}}\Rightarrow(1+1)p\_{\text{LoRA}}.$
>
> **FLoRA:** Each client still uploads its own LoRA parameters, but the server stacks all clients' low-rank modules to form an effective rank-$nr$ adapter and broadcasts the stacked LoRA back to all clients. Therefore, the downlink to each client is n times larger than in FedIT, $p\_{\text{FLoRA}}
> =\underbrace{p\_{\text{LoRA}}}\_{\text{uplink}}+\underbrace{n\cdot p\_{\text{LoRA}}}\_{\text{downlink}}
> = (1+n)\cdot p\_{\text{LoRA}}.$
>
> FFA-LoRA: FFA-LoRA freezes all $A\_\ell$ and only communicates $B\_\ell\in\mathbb{R}^{d\_\ell\times r}$. In terms of order, this reduces the LoRA payload by roughly a factor of two when $\sum\_\ell d\_\ell \approx \sum\_\ell k\_\ell$, leading to $p\_{\text{FFA-LoRA}}\approx\underbrace{\tfrac{1}{2}p\_{\text{LoRA}}}\_{\text{uplink}}+\underbrace{\tfrac{1}{2}p\_{\text{LoRA}}}\_{\text{downlink}}= p\_{\text{LoRA}}.$
>
>  FRLoRA: FRLoRA performs FedAvg on the LoRA parameters (as in FedIT) and communicates a full-size residual per layer to correct aggregation noise. Thus, $p\_{\text{FRLoRA}}=\underbrace{p\_{\text{LoRA}}}\_{\text{uplink}}+\underbrace{p\_{\text{LoRA}}}\_{\text{downlink (LoRA part)}}+\underbrace{p\_{\text{full}}}\_{\text{downlink (dense residual)}}= 2\cdot p\_{\text{LoRA}} + p\_{\text{full}}.$ where the additional $p\_{\text{full}}$ term dominates for large LLMs, making FRLoRA no longer communication-efficient.
>
> For FedMomentum, each client uploads its local LoRA parameters once per round (same as FedIT), i.e., $p\_{\text{LoRA}}$. On the downlink, the server sends back a low-rank reconstruction consisting of the major $r$ components (forming the new LoRA) and an additional residual subspace of rank $s$ (still in low-rank form, not full dense). This corresponds to an effective rank $r+s$ adapter, so the downlink size is $\frac{r+s}{r}\cdot \cdot p\_{\text{LoRA}}$.
>
> For comparison with FLoRA's rank-$nr$ stacking, it is convenient to write $\lambda = \frac{r+s}{nr}$.
> Since the total rank after aggregation is upper-bounded by $nr$, we have $0 \le s \le (n-1)r$, which implies $\lambda = \frac{r+s}{nr} \in \left[\frac{1}{n},1\right].$
>
> Then, the per-round, per-client communication cost of FedMomentum can be written as
> $p\_{\text{FedMomentum}}=\underbrace{p\_{\text{LoRA}}}\_{\text{uplink}}+\underbrace{\lambda n\cdot p\_{\text{LoRA}}}\_{\text{downlink}}= (1+\lambda n)\cdot p\_{\text{LoRA}}.$ In other words:
>
> * When $s=0$ (no residual subspace), $\lambda = 1/n$ and FedMomentum degenerates to the FedIT communication level: $(1+\lambda n)\cdot p\_{\text{LoRA}} = (1+1)\cdot p\_{\text{LoRA}};$
>
> * When $s=(n-1)r$ (maximal residual), $\lambda=1$ and the downlink matches FLoRA's stacked rank-$nr$ adapter: $(1+\lambda n)\cdot p\_{\text{LoRA}} = (1+n)\cdot p\_{\text{LoRA}}.$
>   Therefore, ours achieves lower communication cost than FLoRA while delivering consistently better convergence, and in some cases its overhead is comparable to FedIT, making it a more communication-efficient and effective approach for federated LoRA fine-tuning.

---

> ### Author Response · Authors · 2025-11-24
> **Author response to Reviewer mVza (Part 3 of 3)**
>
> > Q3: sensitivity analysis (LoRA rank r, the residual energy threshold, randomized-SVD oversampling/power iterations)
>
> **For LoRA rank r:** We additionally include a rank-sensitivity study here. We conducted additional experiments on the MATH Reasoning task under rank = 16, 32, and 64 (in the following table). Across all rank settings, FedMomentum consistently achieves the best performance, demonstrating that our method is robust to different LoRA ranks and does not rely on a specific hyperparameter choice.
>
> | Method      | r=16  | r=32  | r=64  |
> | ----------- | ----- | ----- | ----- |
> | FedIT       | 19.18 | 10.72 | 23.81 |
> | FLoRA       | 29.72 | 29.06 | 8.34  |
> | FFA-LoRA    | 21.61 | 25.40 | 7.28  |
> | FRLoRA      | 8.04  | 26.48 | 34.95 |
> | FedMomentum | 32.75 | 34.22 | 35.33 |
>
> **For the residual energy threshold:** As shown in Figure 2, the tail singular values are over two orders of magnitude smaller than the leading ones. Their squared contribution is therefore <0.01% of the total energy, meaning that the top components already capture essentially all useful information in the aggregated update.
>
> Also, considering that SVD-based methods typically regard >99% cumulative energy as sufficient to preserve the matrix structure, we adopt a conservative threshold of 99.99% to avoid introducing any approximation bias during aggregation. This ensures that all meaningful directions are retained, while only removing numerically negligible components.
>
> Formally, we compute $E(t)=\frac{\sum\_{j=1}^t\sigma\_j^2}{\sum\_{j=1}^{nr}\sigma\_j^2},$ and define $r\_{\text{eff}}=\min\{t:E(t)\ge\tau\},\quad s=r\_{\text{eff}}-r,\quad \tau=0.9999.$
>
> Although a smaller threshold would further reduce communication, our priority is avoiding bias and ensuring stable training, and 99.99% achieves this while keeping the residual rank very small in practice.
>
> **For randomized-SVD oversampling/power iterations:** In our approach, we adopt randomized SVD to efficiently approximate the low-rank decomposition of the aggregated update $\Delta W$. As we know the maximum rank of $\Delta W$ is nr, where n is the number of clients and r is the rank for each client's update, we set the sketch size c = nr. This choice ensures that we capture all significant singular values and singular vectors of the matrix without any loss of information, as the rank of \Delta W is bounded by nr.
>
> Since we already know the matrix rank and have chosen c = nr, there is no need for additional oversampling (increasing c > nr) or power iterations. These techniques are typically used to improve accuracy when the rank is unknown or when a larger sketch is required to capture a broader range of singular values. However, in our case, the choice of c = nr suffices to preserve the essential structure of the matrix, making oversampling and power iterations unnecessary.

---

### Author Response · Authors · 2025-11-24
**Official Comment by Authors**

We thank all reviewers for their careful evaluations and constructive feedback. We are encouraged by the positive remarks across different aspects of the work. Reviewer mVza highlighted the clear problem formulation around momentum loss and the principled SVD-based solution, along with the consistent empirical gains. Reviewer rTjD emphasized the improvements on math, commonsense, and code benchmarks and the effectiveness of the balanced $\sqrt{\Sigma}$ split. Reviewer hXhJ acknowledged the consistent performance gains across datasets. We sincerely appreciate these insights and address the raised concerns respectively. If there are any further questions or points requiring clarification, we would be happy to continue the discussion.

---

### Author Response · Authors · 2025-11-27
**Official Comment by Authors**

We respectfully inform the reviewers that we have also uploaded the revised manuscript, with all changes marked for ease of reference. We truly appreciate your constructive feedback and are happy to address any further comments or discussion.

---

### Meta-Review · Area_Chair_6rR4 · 2026-01-05

**Summary:**

This work presents a federated fine-tuning framework for LoRA-based models that aggregates client updates in the multiplication of LoRA-B and LoRA-A space and then applies SVD to the aggregated updates. This work has three reviewers with a positive reviewer (8) and two negative reviewers (4, 4). Although the authors provide many rebuttal contents, two negative reviewers are not positive to raise their scores. After checking the rebuttals, the rebuttal texts can not fully change the scores of the negative reviewers. Hence, this work can not be accepted by ICLR 2026.

**Reviewer Concerns:**

Main concerns:
1. incremental contribution of the proposed FedMomentum due to the high similarity to the cloest work FlexLoRA.
2. more ablation study on key knobs
3. “Momentum” story isn’t operationalized

**Reviewer Scores:**

This work has three reviewers with a positive reviewer (8) and two negative reviewers (4, 4). No reviewers change their scores.

---

### Decision · Program_Chairs · 2026-01-26

Reject